# Three Techniques for Enhancing Chaos-Based Joint Compression and Encryption Schemes

**DOI:** 10.3390/e21010040

**Published:** 2019-01-09

**Authors:** Chao-Jen Tsai, Huan-Chih Wang, Ja-Ling Wu

**Affiliations:** 1Department of Computer Science and Information Engineering, National Taiwan University, Taipei 106, Taiwan; 2Graduate Institute of Networking and Multimedia and Department of Computer Science and Information Engineering, National Taiwan University, Taipei 106, Taiwan

**Keywords:** compression, encryption, chaotic map, look-up table, streaming data

## Abstract

In this work, three techniques for enhancing various chaos-based joint compression and encryption (JCAE) schemes are proposed. They respectively improved the execution time, compression ratio, and estimation accuracy of three different chaos-based JCAE schemes. The first uses auxiliary data structures to significantly accelerate an existing chaos-based JCAE scheme. The second solves the problem of huge multidimensional lookup table overheads by sieving out a small number of important sub-tables. The third increases the accuracy of frequency distribution estimations, used for compressing streaming data, by weighting symbols in the plaintext stream according to their positions in the stream. Finally, two modified JCAE schemes leveraging the above three techniques are obtained, one applicable to static files and the other working for streaming data. Experimental results show that the proposed schemes do run faster and generate smaller files than existing JCAE schemes, which verified the effectiveness of the three newly proposed techniques.

## 1. Introduction

In recent decades, the amount and size of multimedia files as well as the needs for secure transmission of confidential information over public networks are on the rise. In addition, with the popularity of the Internet of Things (IoTs), more and more devices with limited batteries are asked to securely transfer data over the Internet. Therefore, supporting both the functions of compression and encryption becomes a common requirement in most of today’s data transmission systems. This explains why a large number of recent signal processing tasks properly combined the above two operations together, so as to provide better security, less communication cost and lower energy consumption, at the same time. Moreover, time critical and privacy sensitive multimedia telecommunication services, such as voice chatting, video messaging, videoconferencing, video hosting and streaming (e.g., YouTube and Twitch) are becoming increasingly popular and pervasive. Consequently, there is a strong need to develop effective and efficient JCAE schemes.

In general, there are two different directions in developing JCAE schemes. One is to embed key-controlled confusion and diffusion mechanisms into compression algorithms, the other is to introduce compression function into cryptographic schemes. The first direction has attracted more attention in early years, because the outcomes of entropy coding algorithms such as Huffman codes and arithmetic codes are easily turned to ciphers by using a secret key to govern the statistical model used in encoding. The receiver is able to synchronize the changed statistical model and produce the correct output only when the secret key is provided, correctly [1,2,3].

Works on embedding encryption into source-coding schemes focused mainly on the manipulation of interval allocations using a secret key [1,2]. For example, in Huffman coding, a multiple Huffman table-based approach [3] has been proposed to swap the left and the right branches of an intermediate node, according to a selected secret key, in a Huffman tree. As for arithmetic coding, approaches such as randomized coding [4] and interval splitting [5] have also been investigated.

The prominent properties of chaotic systems, such as sensitive to initial conditions, topologically mixing, and having dense periodic orbits, are ideal for building cryptography schemes [6]. Moreover, chaotic maps are iterative formulas, which in comparison to continuous time chaotic systems, are not required to establish the correct step size and numerical method to implement them in digital hardware, as demonstrated in [7]. Hence, there is an increasing trend of designing JCAE schemes based on chaotic maps. A chaos-based cipher using a key-dependent chaotic trajectory was designed by Baptista [8]; however, it suffers from the shortcoming of huge ciphertext expansion. Attempts were made in [9,10,11,12,13] to incorporate certain compression capabilities into the Baptista-type chaos-based cipher systems.

In this work, three techniques are proposed to further enhance the Baptista-type JCAE schemes. The first accelerates the scheme of [10] by using two additional auxiliary data structures to avoid modifying the lookup table used for encryption. The second extends the idea of [12] and improves the corresponding compression ratio. It selects a small set of “Important Trigrams” for being used along with the “Bigrams” adopted in the original scheme. This small set of selected “Trigrams” adds little storage overhead but impacts significantly on the compression performance. The last is a space efficient adaptive probability estimation method, which is suitable for modeling streaming data. In the proposed model, we exponentially weight the plaintext symbols in the stream and take the weighted average occurrence probability as an estimate. In contrast to the method presented in [13], our model retains information with respect to older symbols in the stream. As a result, it provides a more accurate estimate of the occurrence probability distribution. Finally, two effective and efficient chaos-based JCAE schemes, in which the proposed techniques have been embedded, one for static files and the other for streaming data are proposed.

This paper is organized as follows: in Section 2 some related chaotic-based JCAE schemes are reviewed. Section 3 addresses and details the three proposed techniques and the two JCAE schemes. Section 4 presents the experimental results and the security analyses of the proposed approaches. Lastly, conclusions are drawn.

## 2. Related Work

### 2.1. Look-Up Table and Chaotic-Map Based Approach

An efficient chaos-based cipher, proposed in [9], searches for the target plaintext symbol through a codeword-based look-up table (LUT) using a key-dependent chaotic trajectory and treats the required number of iterations of the chaotic map as the ciphertext. The LUT used for encryption is constructed according to the probabilities of occurrence of all plaintext symbols. The ciphertext is then compressed by entropy coding and the compressed ciphertext stream is masked by a pseudo-random bitstream to further enhance the security.

The LUT is constructed by dividing the phase space of the chaotic map into a number of equal-width partitions, and the number of partitions mapped to a particular symbol is proportional to its probability of occurrence. The partitions are pseudo-randomly permuted so that the partitions that are mapped to the same symbol spread over the whole phase space of the chaotic map.

To avoid large number of iterations, only source symbols with higher probabilities of occurrence are encrypted by searching through the LUT, meanwhile the less probable symbols with lower probabilities are left intact, and are only encrypted by the last masking step. We call the symbols encrypted by searching through the LUT working in the “search” mode, while the others in the “mask” mode. As the symbols working under the mask mode do not use the LUT, they are ignored during the LUT construction stage. During the LUT construction, an additional “specific” symbol is needed to indicate that the next symbol is working under the mask mode.

In search mode, the target plaintext symbol is searched within the LUT. The corresponding ciphertext is defined to be the number of iterations of the chaotic map required until the chaotic trajectory lands on a partition in the LUT which is mapped to the target symbol. This searching process is analogous to randomly drawing a symbol from the set of plaintexts until the target plaintext symbol is found.

A trick for speeding up, introduced in [9], is to split the LUT into a few sub-LUTs. Original plaintext symbols are contained in at most one sub-LUT. The search mode uses the sub-LUT containing the target symbol to do encryption. If the LUT is split into *n* sub-LUTs, it was called an n-map approach in [9].

### 2.2. Dynamic Updating Look-Up Table Based Approach

To improve the compression performance, in [10], the LUT used for encryption is dynamically updated in the searching mode. The same as [9], the number of iterations of the chaotic map used for finding the target symbol is defined as the corresponding ciphertext. If the partition the chaotic trajectory landed on maps to a non-target symbol, all partitions associated with the visited non-target symbol are relabeled to the non-visited symbol with the highest probability of occurrence. Since the number of partitions labeled with the target symbol gradually increases, fewer iterations are required for encryption, and therefore, shorter ciphertext can be expected.

### 2.3. Number of Distinct Plaintext Symbol Based Approach

Basically, the encoding process described in [11] is identical with that of [9], except that the ciphertext is defined to be the number of distinct symbols visited until the target symbol is hit.

### 2.4. Bi-Gram Based Approach

In [12], the authors constructed multiple sub-LUTs based on the Conditional Bigram Probability (CBP) of two consecutive source symbols occurring in the input to build their scheme. The Normalized Conditional Bigram Probability (NCBP) of two consecutive symbols $s_{i}x$ is denoted as:$$P\left(S_{i}x|s_{i}\right)=\frac{P\left(s_{i}x\cap s_{i}\right)}{P\left(s_{i}\right)}=\frac{P\left(s_{i}x\right)}{P\left(s_{i}\right)}$$
where $S=\left\{s_{1},s_{2},\ldots ,S_{N}\right\}$ denotes the plaintext alphabet set and x∈S. The whole bigram set is parsed into bigram subsets starting with different leading symbols, and then each subset constructs its own conditional sub-LUT according to the NCBPs of its set members.

Instead of using the same LUT to encrypt every input symbol, [12] outperforms [11] by using the conditional sub-LUT corresponding to the just-before-being-encrypted symbol for encryption. Since the first symbol does not have a preceding symbol, it uses the same LUT as that of [11]. Simulation results showed that bigram based LUT performs better than its unigram counterpart, in terms of compression ratio.

### 2.5. JACE Scheme for Streaming Data

The approach presented in [13] dynamically updates the LUTs during the searching process in [12]. This dynamic updating mechanism removes the overhead of storing numerous LUTs needed in [12] and takes advantages of local features of the plaintext to provide better compression ratio, for dealing with streaming data.

As a modification of [12], the streaming scheme proposed in [13] also uses NCBPs. Real-number LUTs, which can be easily and efficiently tuned, are included. Each bigram subset keeps track of the ***W*** last bigrams in the subset that occurred in the input stream. The real-number LUT corresponding to the subset is updated per plaintext symbol, according to the NCBP of the last ***W*** bigrams of the subset. Finally, the Huffman coding step used in [12] is replaced by dynamic Huffman coding presented in [13]. Notably, the dynamic nature of this scheme makes it possible to process the input stream in only one pass.

## 3. The Proposed Techniques for Performance Enhancement

We first present three new techniques that respectively improve the JCAE schemes presented in [10,11,12,13]. They served also as building blocks for constructing complicated JCAE schemes. As an example, a static JCAE scheme and a streaming JCAE scheme that leverage the three newly proposed techniques will be addressed in Section 3.4 and Section 3.5, respectively.

### 3.1. A Queue Based Non-Relabeling Scheme for Look-Up Table Updating

References [10,11] took different approaches to improve the compression ratio of [9]. The former provides a better compression ratio, but its high computational cost prevents it from being used in practice. Our scheme is the same as that of [10] except that it is accelerated using a new algorithm, making our approach the fastest and most preferable of the three.

For convenience, we say a symbol is visited if a partition labeled with it has been hit by the output of a chaotic map. In [10], it was observed that all partitions with a visited non-target symbol are relabeled to the same symbol, namely, the current most probable non-visited symbol (MPNVS). This is because the MPNVS changes only when the last MPNVS is visited, and when it is visited all partitions labeled with the last MPNVS are relabeled to a new one.

The performance bottleneck of [10] comes from the time spent on LUT updating. Based on the previous observation, instead of relabeling the partitions of an LUT, we maintain two additional auxiliary data structures: One is a timestamp table which keeps track of the visited symbols and the other is a queue whose front pointing at the current MPNVS.

#### 3.1.1. The Timestamp Table

The timestamp table is used to check if a symbol has been visited. By using timestamps instead of Boolean values, we avoid initializing the table before encrypting every single symbol. The timestamp table stores the timestamps of symbols in an LUT. A symbol’s timestamp records the last “time” it was visited, where the “time” stands for the position of the source symbol in the input file. For example, if a symbol was latest visited when encrypting the 4th symbol, its timestamp would be 4. When a symbol is visited, its timestamp in the table is updated to the current time. To see if a symbol has been visited or not, we just examine the symbol’s entry in the timestamp table and compare the recorded timestamp with the current one.

#### 3.1.2. The MPNVS Queue

The MPNVS queue contains the symbols that appeared in an LUT, sorted according to their occurrence probabilities from high to low. The MPNVS queue is dequeued whenever its front is (pointed to) a visited symbol. This mechanism guarantees that the front is always an MPNVS because the symbols are ordered by the magnitude of occurrence probability. The queue is restored before encrypting the next symbol. Although it is conceptually a queue, the MPNVS queue is easier to be implemented as an array, with a pointer pointing to the queue’s front. Dequeuing is executed by incrementing the pointer and restoring the queue is executed by pointing the pointer to the head of the array. All operations, including restoring the queue, take θ(1) time, only.

#### 3.1.3. The Proposed Scheme

The above two structures collaborate to compute the symbol that will be compared with the source symbol. We call the computed symbol CS. The encryption procedure can be addressed as follows:(1)Increment the current time and restore the MPNVS queue.(2)Use the chaotic map to sample a partition in the LUT. Let CS be the partition’s label.(3)Query the timestamp table to know if CS has been visited. If it hasn’t, skip and go to step-5.(4)Find out the MPNVS by dequeuing the MPNVS queue until the front symbol has not been visited. Set CS to the MPNVS.(5)Update CS’s timestamp table entry to the current time.(6)Compare CS with the source symbol. If they are different, return to step-2.(7)Output the number of chaotic iterations as the corresponding ciphertext.

#### 3.1.4. Time Complexity Analysis

Our scheme takes θ(|A|) time to initialize, where |A| denotes the alphabet size. It is ignorable since the time spent on constructing the LUTs for all three schemes is Ω(|A|). The time complexities for encrypting a symbol are shown in Table 1; clearly, our scheme is orders faster than that of [10,11].

### 3.2. A Hybrid Unigram and Bigram Context Based Look-Up Table Selection Mechanism

Reference [12] improved the compression ratio of [11] by using multiple LUTs based on CBPs of two consecutive source symbols in the input. In other words, it uses the preceding symbol (a unigram) as context to predict the current symbol. Every context has its own LUT, which is constructed by gathering statistics of symbols in that context. The encoder uses a symbol’s context as additional information to aid the compression and encryption. In their experiments, better modelling of the source data resulted in increased prediction accuracy, which in turn improved both the compression ratio and the execution time.

On observing the success of unigram context model, one would extend the approach by using bigram context model; in other words, using the two previous symbols for prediction. If we compare the files compressed using unigram contexts and bigram contexts, the bigram context compressed files are considerably smaller if we exclude the costs of LUTs’ sizes from counting; however, if we include them, the unigram context compressed ones become smaller in size. This fact shows that the additional information the bigram contexts provided is useful for compression, but the overhead required to store the information itself is not negligible. As a result, both [12,13] used only unigram contexts.

The Pareto principle states that for many events roughly 80% of the effects come from 20% of the causes. Applying the gist of it to bigram contexts, most of the compression gain is the result of a handful of bigram contexts, only. Therefore, to provide both higher compression gain and smaller storage cost, we should sieve only those bigram contexts out which are useful for compression and ignore the others. In other words, we need not to store the LUTs for storing useless bigram contexts. But, the question is “how to sieve the so-called useful bigrams out?”. In the following paragraphs, we first explain how the bigram and unigram contexts are utilized, and then a simple guideline for selecting the appropriate bigrams is provided.

#### 3.2.1. Hybrid Bigram and Unigram Contexts

For convenience, we say that an *n*-gram is useful if the context corresponding to it is useful. A unigram is always useful. We also call the LUT of an *n*-gram context an *n*-gram-LUT. For example, a bigram-LUT is the LUT of a bigram context, not a table containing bigrams.

Suppose we already have the set of useful bigrams UB. The useful bigram and trigram LUTs are created by analyzing the input file or the streaming data. When encoding a plaintext symbol, we look at both the bigram and the unigram preceding it. For example, for the “C” in “ABC”, the bigram is “AB” and the corresponding unigram is “B”. If the preceding bigram is useful then we use the bigram-LUT; otherwise, we use the unigram LUT as a fallback.

#### 3.2.2. Bigram Selection

Assume we already have the bigram and trigram distributions of the plaintext file, ideally, we could calculate the expected number of bits saved by using bigram contexts and then choose those that are beneficial. Unfortunately, this is impossible due to the difficulty of calculating the ciphertext distribution, of [10,11], in advance. As a result, we can only rely on heuristics to choose useful bigram contexts.

A rule of thumb is to pick the bigrams that occurred most frequent in the plaintext. They will be used the most if employed, and a larger sample size implies a more accurate distribution estimation. We tried to take other factors into account, but the results were not pleasing.

The next thing to decide is the number of bigram contexts used. Plotting the bigram histograms of a few testing files, we observed that the bigram distributions are often concentrated at a small range of numbers. If we order the bigrams according to the frequencies of occurrence, from high to low, the graph is extremely skewed towards the high frequency side, and for the majority of other rare-occurred bigrams the graph is nearly a horizontal straight line, meaning that their frequencies are roughly the same. The graph somewhat resembles a plot of the function y=1/x except being much more skewed. On the basis of the above observation, despite quoting the Pareto principle (a.k.a.80-20 law) as our motivation, we actually select far less than 20% of the bigram contexts as the useful ones. In practice, we suggest using 256 bigram contexts, which are usually enough to cover the high probability parts of the bigram histogram. This is certainly not the optimal choice for every case, but it would be good enough for normal usages.

### 3.3. An Adaptive Probability Estimation Modeling for Streaming Data

The statistic model adopted in [13] is derived from a sliding window of size ***W***. It estimates the probability distribution of the current symbol by storing the previous ***W*** encountered symbols in a queue and maintaining the histogram of the queue. New symbols are enqueued, and when the queue’s size exceeds ***W*** the oldest symbol is dequeued.

Adaptive schemes often need to maintain multiple statistic models. For example, the streaming scheme in [13] maintains a model for each symbol in the plaintext alphabet. In such cases, the model size becomes a heavy burden if the alphabetical size of the plaintext is large. We try to use a new modelling methodology with a much more compact representation so that we can put most of the chosen models into the CPU cache.

Another downside of [13]’s modelling is that it discards all information of a symbol once it leaves the sliding window. Under some circumstances this is preferred because it implies a stronger dependency on recent symbols, but in [13]’s streaming scheme, according to their experiment results, this is not always the case.

To fix this shortage, an adaptive model suitable for estimating the probability distribution of streaming data, is proposed in the following. It uses less memory and retains the information of symbols that appeared a long time ago, making it provides more accurate statistical estimation for streaming schemes.

#### 3.3.1. Retaining Old Information

Our new modelling consists of a frequency distribution estimation DE, and a “box” B of size ***W***. Initially the distribution is assumed to be uniform, and the box is empty. When a symbol is fed as an input to the model, it is put into the box. When the box is full, i.e., it contains ***W*** symbols, we update DE by averaging it with the frequency distributions of symbols in the box (called DB in the rest of this work). That is, the newly estimated probability of a symbol is the average of its estimated probabilities DB and DE.

The above process can be treated as taking a weighted average of all symbols fed into the queue (or model). By taking average of DB and DE, we essentially halve the weights of symbols fed ***W*** symbols ago. A symbol fed ***2W*** symbols ago would have a quarter of the current symbol’s weight. By this way, we manage to preserve information of old symbols while adapting to new ones.

#### 3.3.2. Implementation Details

The model described above is implemented using an integer weight array, ***Weight***, and a counter that counts how many symbols have been fed into the model. The array maps a plaintext symbol to a number directly proportional to the estimated probability of that symbol. That is, the estimated probability of a symbol *S* is:$$\tilde{P}\left(S\right)=\frac{Weight\left(S\right)}{\sum_{C}Weight\left(C\right)}$$

Elements of the array are initialized to W/|A|, and the counter is initialized to 0. When the model is fed an input symbol, we increment both its entry in the array and the counter by 1. This corresponds to putting the symbol into the box B.

If the counter is a multiple of ***W***, the box is full. We maintain the property that the array sums to ***W*** after emptying the box, so when the box is full again the array sums to ***2W***. At this point, the array can be viewed as the sum of the old estimated frequency distribution and the distribution of the most recent ***W*** symbols, which respectively corresponds to DE and DB.

When the box is full, we rescale ***Weight*** by halving the values of all entries. Remember that ***Weight*** stores integers, so it is possible that the value doesn’t sum to ***W*** due to the division of odd integers by 2. In order to maintain the property mentioned in last paragraph, we add 1s to random entries in ***Weight*** until it sums to ***W***. Since, typically, large ***W***s are used, randomly adding a few 1s here and there doesn’t significantly affect the prediction accuracy.

This implementation eliminates the need of storing the previous symbols, so it takes much less memory space than the adaptive model used in [13]. Notice that we can store each element in the ***Weight*** array in 2 bytes as long as ***W*** doesn’t exceed **2^15^**.

### 3.4. The Proposed Static Scheme

The block diagram of the proposed static JCAE scheme is shown in Figure 1. It is based on the scheme in [12], but with three major and one minor modifications. In this section, ***N*** denotes the size of the symbol’s alphabet set.

The first modification is to employ the method, described in Section 3.2, for selecting and utilizing the 256 additional bigram context LUTs along with the ***N*** unigram context LUTs that [13] used. This includes modifying the LUT creation step and the chaos-based encryption step.

The second modification is to replace the original LUTs with the ones using the scheme proposed in Section 3.1. Moreover, a masking step is added after the Huffman coding step. Notice that this is not the mask mode described in Section 2. Our masking step uses the method described in [9] to diffuse the differences between the plaintext stream and the earlier parts of the encrypted file. We will describe the process in more details later.

Since the initial LUT in [12] is used only once, creating the LUT and storing it in memory seems to be a waste, so we try to avoid doing so. Alternatively, we use one of the unigram or bigram LUTs to encrypt the first symbol. We first parse the file to create unigram and bigram LUTs using the technique presented in Section 3.2, and then encrypt the plaintext stream using the scheme addressed in Section 3.1. The encrypted ciphertext is compressed using the Huffman coding, and finally, the compressed data is masked to boost the security.

As remarked by one of the reviewers, the masking module (highlighted in Figure 1) has demonstrated to be a good option in encryption schemes implemented in Field-Programmable Gate Arrays (FPGAs), see [14] for details.

#### 3.4.1. LUT Creation

The whole plaintext sequence is scanned once to compute the bigram and the trigram distributions. The bigram distribution is used to both create the ***N*** unigram context LUTs and compute the 256 usable bigram contexts, as stated in Section 3.2. The trigram distribution is then used to construct the 256 bigram LUTs. All LUTs created use the scheme addressed in Section 3.1. The LUTs also use the 16-map approach described in [9], which splits an LUT into multiple sub-tables and assigns symbols to a sub-table according to their frequencies of occurrence. The LUTs should then be sent to the decoder.

#### 3.4.2. Chaos-Based Encryption

On the one hand, we keep track of the last two plaintext symbols stored in memory for selecting the appropriate LUT used to encrypt the current symbol. On the other hand, the first two symbols in memory are encrypted using a random LUT chosen also by applying a chaotic map. After that, the plaintext symbols are sequentially encrypted using the aforecited last two symbols as context to choose the LUT for doing encryption. The bigram LUT is preferred over the unigram LUT, in general. Once the appropriate LUT is determined, we apply the scheme presented in Section 3.1 to encrypt the symbol.

The mask mode addressed in [9] is also used. If a symbol is not in the chosen LUT then its processing is transferred to mask mode. In mask mode, a symbol is encrypted simply by XORing the symbol with a pseudo random byte. Of course, as in [9], an additional mask-mode symbol is introduced to indicate that the next symbol is transferred to mask mode.

Every time a symbol is encrypted, we extract 8 bits of the chaotic map’s parameter $x_{i}$ and appended it to the mask sequence. After this step, the mask sequence will be as large as the plaintext file. This sequence is used in the 4th step of Figure 1 to mask the compressed ciphertext.

#### 3.4.3. Huffman Coding

As pre-described, the chaos-based ciphertext is compressed using Huffman codes. We call the compressed stream of symbols the intermediate sequence. Symbols encrypted in mask mode are not compressed using Huffman codes, but the additional mask-mode symbol is used instead.

#### 3.4.4. Masking

Both the intermediate sequence (obtained from the 3rd step of Figure 1) and the mask sequence (obtained from the 2nd step) are divided into 32-bit blocks indexed from 0. The ciphertext for the *i*-th block is given by:$$c_{i}=\left(r_{i}+c_{i-1}+m_{c_{i-1}+r_{i+1}\;mod\;L}\right)\;mod\;2^{32}$$
where $r_{i}$ and $m_{i}$ are respectively the *i*-th blocks in the intermediate sequence and the mask sequence, and *L* is the number of blocks in the mask sequence.

The intermediate sequence should be masked twice in order to propagate the changes at the tail of it to the whole sequence.

This step requires a 32-bit block $c_{-1}$ from the secret key. The authors of [9] suggested that the block $r_{L}$, required to decrypt the last block $c_{L-1}$ in the ciphertext sequence, should be set to the same value as $c_{-1}$.

### 3.5. The Proposed Streaming Scheme

The block diagram of the proposed streaming scheme is shown in Figure 2. It is exactly the same as that given in [13]. Indeed, our scheme is based on the approach presented in [13]; however, all three newly proposed techniques are applied to vastly enhance the corresponding compression ratio and execution speed. The LUT scheme given in Section 3.1 is adopted; nevertheless, instead of using real-number LUTs, 256 selected bigram LUTs are included and the original sliding window model is replaced by the model proposed in Section 3.3.

The core of the streaming scheme is the LUT updating step, which actually includes adaptively choosing bigram context models and the modelling for each context. The chaos-based encryption is the same as that of the static scheme. Notice that the well-known Vitter’s algorithm [15] is used for conducting the dynamic Huffman coding step.

#### 3.5.1. The Issue of Real-Number LUTs

In the streaming scheme given [13], real-number LUTs are used instead of integer-partitioned ones [11]. A real-number LUT resides in the interval [0, 1) and is divided into multiple sub-intervals proportional to the symbols’ occurrence frequencies. Recalling from the interval and interval splitting methods commonly used in Arithmetic Coding (AC), it follows that Real-number LUTs behave almost the same as that of AC, except that the sub-intervals of it must be ordered from broad to narrow, i.e., the symbols are ordered from high to low in terms of frequencies.

The reason for choosing real-number LUTs in [13] is owing to the ease of their updating. real-number LUTs can be updated per plaintext symbol, which is nearly impossible for the other LUT schemes. However, doing real-number LUT’s update sacrifices a lot of security. Because the symbols are ordered, one end of the table always stores the most frequent symbol and the other end must be the least one. This ordered table partitioning itself leaks some information. What’s worse is that the sub-intervals used for representing symbols are continuous, unlike the other Bapista-based LUTs where intervals corresponding to symbols are scattered over the table. Having continuous intervals makes it much easier to reconstruct a real-number LUT by launching probing attack repeatedly.

Owing to the above security concerns, the LUT scheme proposed in Section 3.1 is adopted in place of the real-number one. Our LUT scheme would be much more secure; however, in this situation, the only way to update an LUT is to reconstruct it again, so our streaming scheme cannot afford to update the LUTs as frequent as that of [13].

#### 3.5.2. Context Modelling

The adaptive model, described in Section 3.3, is used to model the contexts of streaming data. The initial estimated distribution is the uniform one since we have no information about the plaintext stream yet. As more symbols are fed into the contexts, the model learns the characteristics of the input stream and becomes more and more accurate.

When a bigram is useful we create a model and an LUT for it. The model is copied from the unigram model and the LUT is randomly permuted using a chaotic map. The model is updated if the bigram remains useful, and every time the sliding window is full we reconstruct the corresponding LUT again. When an LUT becomes useless, its model and LUT are discarded. When both the bigram context and the unigram context are present for the encoded symbol, we update both context models although only the bigram context is used for prediction.

#### 3.5.3. Bigram Context Selection

Applying the technique presented in Section 3.2, we use 256 additional bigram contexts on top of the unigram contexts. Different to the static scheme, at the beginning we don’t know which bigrams are useful. As a result, we need to adaptively update the useful bigram list and replace bigram contexts that have become useless with those that perform better.

In Section 3.2 we use the bigram frequency as a heuristic to decide which bigrams are useful. We use the same heuristic in this scheme. Reasonable estimations of bigram frequencies are obtained by using the unigram models proposed in Section 3.3. A unigram model is based on the number of occurrences of the unigram in the encoded plaintext. If A and B are symbols, we obtain the estimated bigram probability of AB by:$$\frac{Counter_{A}}{L}\cdot \tilde{P_{A}}\left(B\right)=P\left(A\right)\cdot \tilde{P}\left(B|A\right)=\tilde{P}\left(AB\right)$$
where L is the total number of plaintext symbols already encoded, $Counter_{A}$ is the counter of A’s model, $\tilde{P}\left(\cdot \right)$ is the estimated probability function and $\tilde{P_{A}}\left(B\right)$ is the estimated probability of B using A’s model.

We maintain a list of useful bigrams. Notice that the bigrams in this list are not necessarily the 256 bigrams with the highest estimated probabilities. A bigram in the useful list is replaced only when the new bigram performs much better, preventing bigrams models from having too short a lifespan to adapt to anything. In our experiments, we only replace an old bigram if the new bigram’s probability is more than twice of the old one. According to our experiments, this approach provides better speed and compression ratio than directly using the 256 highest estimated probability bigrams.

#### 3.5.4. The Proposed Procedure

(1)Fill the useful bigram list with 256 random bigrams. Create the models and LUTs for them and the unigrams.(2)Use the contexts and their LUTs to encode the next symbol.(3)Update the bigram model if it exists.(4)Update the unigram model. If the model’s box is not full, return to step 2.(5)For all bigrams starting with the unigram (e.g., AB starts with an A), see if the bigram can replace the worst behaved bigram in the useful bigram list.(6)Rescale the model and return to step 2.

## 4. Performance and Security Analyses

We use the Silesia compression corpus [16] to test the two proposed schemes. It contains 12 files with sizes between 6 MB and 51 MB, including two medical images, the SAO star catalog and a few executable files. The logistic map [17] is chosen as the underlying chaotic map. The plaintext symbols are structured in bytes, i.e., all schemes use 16-maps. The proposed static scheme is benchmarked with the static schemes in [12,13].

The streaming scheme is benchmarked with [13]’s streaming scheme. In [13], the multi window approach with a window size of 512 is used, which yielded the best results. For our scheme, we also use a window size of 512. The performance comparisons of the static schemes and the streaming schemes are respectively shown in Table 2 and Table 3.

### 4.1. Compression Ratio

In this work, the compression ratio is computed as follows:$$\frac{Ciphertext\;Length}{Plaintext\;Length}\times 100\%$$

In all three static schemes, the sizes of LUTs are included in the size of the ciphertext file. It can be seen that our scheme provides significant compression boosts on a large portion of the files, and in the worst case the file size is just barely larger. It is almost the same case for the streaming scheme, except that this time there are three files that are marginally larger than those of [13]’s streaming scheme.

### 4.2. Execution Time

The implementations of [12,13] are in C, and ours are in C++. The experiments were run on a 2015-year MacBook Pro Retina with a 2.7 GHz i5 CPU and 8 GB RAM. The execution time is measured in seconds. Our static scheme takes much less time than [12,13] despite generating 256 additional bigram context LUTS and requiring an addition masking step. This shows the proposed LUT’s updating scheme in Section 3.1 is indeed significantly faster than [11]. The execution time of our streaming scheme is nearly half of that of [13].

### 4.3. Security Analysis

#### 4.3.1. Key Space

A logistic map has two parameters *b* and *X*_0_. They are stored in the IEEE double-precision binary floating-point format, which has 1 sign bit, 11 exponent bits and 52 mantissa bits. The sign and exponents are the same for all valid values of *b* and *X*_0_, so the key size of a logistic map is 2 × 52 bits.

The key space of the streaming scheme is 104 ×
*N* bits, where *N* is the number of logistic map used. The static scheme key space includes an additional 32-bit cipher block value c_−1_, so the key space is 104·*N* + 32 bits. It is suggested to use at least two different logistic maps.

#### 4.3.2. Key Sensitivity

The key sensitivity is measured by slightly modifying one parameter at a time and calculating the percentage of bits changed in the ciphertext. The first two columns in the tables are the results of altering *b* and *X*_0_ by 10^−9^, individually.

Our experimental results of static and stream schemes are shown in Table 4 and Table 5, respectively.

As we can see from the two tables, the percentage of bits flipped are all near 50%, proving that our schemes properly utilize the chaotic properties of the logistic maps for secure encryption.

#### 4.3.3. Plaintext Sensitivity

Next, a bit at different position in the plaintext file is flipped, and again we calculate the percentage of bits changed. Notice that in the streaming scheme we only compare the sequences starting from the ciphertext corresponding to the position we changed. This is because streaming scenarios usually neither require nor expect later data in the stream to affect the ciphertext of data already encrypted.

As shown in Table 4 and Table 5, the percentage of bits changed are all close to 50%, showing that the previous encrypted symbols immediately affect the behavior of the current scheme, meaning that our streaming scheme is extreme responsive and also secure.

## 5. Conclusions

Three techniques for improving chaotic joint compression and encryption schemes are proposed. One improves the speed of a Bapista-type chaotic system. The resulting scheme is faster than all other Bapista-type chaotic systems that we knew. The second technique enhances the compression performance of multi-LUT schemes by using a small set of bigram contexts in additional to unigram contexts. The technique provides a substantial compression performance gain while requiring only a small space overhead. The last technique is a space efficient modelling method that is suited for modelling streaming data.

With the aids of the above-mentioned techniques, two new joint compression and encryption schemes are proposed. Through a series of experiments on well-known benchmark datasets, the efficiency (in terms of execution speed), the effectiveness (in terms of compression ratio) and the security (in terms of bit-change sensitivity) of the proposed schemes are justified.

With the increased availability of smart mobile devices and the popularity of internet, considerable attention has been paid to the secure and efficient transmission of high-dimensional information, such as images and videos. Because of their cryptographic-favored characteristics, various chaotic maps haven been applied to developed encryption systems for images [18,19]. Therefore, how to design secure and efficient JCAE schemes for high-dimensional media data is one of our future research direction, for sure. Of course, as suggested by one of the reviewers, finding the changes of our system security measures and overall system performances if our Logistic-map based module were replaced by other pseudorandom number generator modules based on different Chaotic-maps, such as the ones used in [20,21,22,23], is another worthy of exploring research topic. Finally, as remarked by one of the reviewers, we should be aware of the recent progress in hardware implementations of chaotic maps, whose speed is definitely higher than software realizations adopted in this work. For example, the integrated designs of chaotic oscillators [24], which is very useful in the field of Internet of Things (IoTs) due to its very low power consumption and the suitability to be embedded into any system because it is realized in the scale of nanometers; most importantly, the oscillators can be optimized to provide better security, see [24] for details.

## Figures and Tables

**Figure 1 entropy-21-00040-f001:**
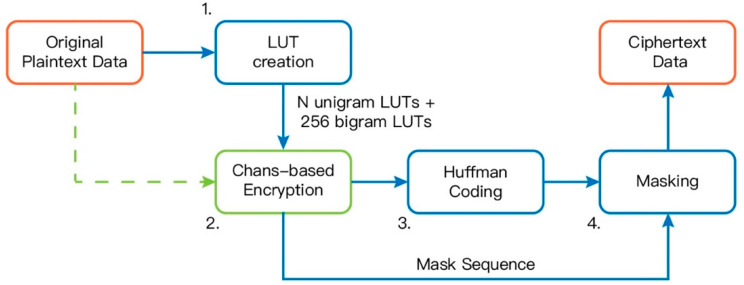
The block diagram of the proposed static scheme.

**Figure 2 entropy-21-00040-f002:**
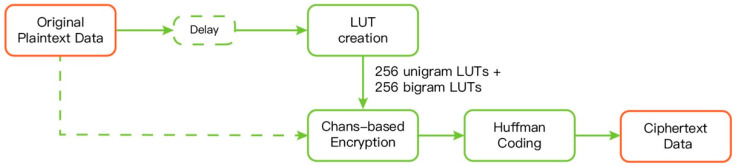
The block diagram of the proposed streaming scheme.

**Table 1 entropy-21-00040-t001:** The time complexity of a single symbol encryption, where k denotes the number of chaotic iterations.

Scheme	[10]	[11]	Our Scheme
**Complexity**	θ(|A|·k)	θ(|A|+k)	θ(k)

**Table 2 entropy-21-00040-t002:** Performance comparisons of the static schemes.

File Name	Reference					
	[12]		[13]		Proposed	
	Ratio	Time	Ratio	Time	Ratio	Time
Xml	49.75%	1.371	49.71%	1.109	40.69%	0.313
Ooffice	71.88%	1.664	71.52%	1.323	68.60%	0.617
Reymont	43.31%	1.720	44.49%	1.244	33.99%	0.372
Sao	83.45%	1.677	83.50%	1.468	77.91%	0.849
X-ray	84.19%	2.153	83.84%	1.756	80.37%	0.833
Mr	49.71%	2.276	51.40%	1.483	40.66%	0.577
Osdb	71.58%	3.288	71.58%	2.267	72.58%	1.093
Dickens	49.88%	3.360	50.09%	2.164	43.21%	0.645
Samba	60.76%	6.918	61.28%	4.496	55.86%	1.714
Nci	31.59%	11.769	33.84%	4.550	21.53%	1.404

**Table 3 entropy-21-00040-t003:** Performance comparisons of the streaming schemes.

File Name	Reference			
	[13]		Proposed	
	Ratio	Time	Ratio	Time
Xml	42.41%	1.213	36.02%	0.645
Ooffice	67.66%	1.785	68.53%	1.104
Reymont	44.60%	1.548	34.73%	0.733
Sao	80.14%	2.095	80.57%	1.296
X-ray	80.00%	2.465	81.09%	1.300
Mr	50.99%	1.958	43.97%	1.522
Osdb	72.57%	3.137	68.57%	1.605
Dickens	50.00%	2.681	42.50%	1.310
Samba	34.42%	5.290	50.88%	2.761
Nci	49.08%	6.182	22.41%	3.500

**Table 4 entropy-21-00040-t004:** Sensitivity of the proposed static scheme.

File Name	Chaotic Map		Plaintext		
	b	*X* _0_	0.25	0.5	0.75
Xml	50.03%	50.00%	50.00%	50.02%	49.99%
Ooffice	50.01%	49.99%	49.98%	49.99%	49.99%
Reymont	50.00%	50.01%	50.02%	50.01%	50.00%
Sao	50.00%	50.01%	50.00%	50.01%	49.99%
X-ray	50.00%	50.00%	49.99%	49.99%	50.01%
Mr	50.01%	50.00%	50.00%	50.00%	50.00%
Osdb	50.00%	50.01%	50.00%	49.99%	50.00%
Dickens	50.00%	50.00%	50.00%	50.01%	50.00%
Samba	50.00%	50.00%	50.00%	50.00%	50.00%
Nci	50.00%	50.00%	50.00%	50.00%	50.01%

**Table 5 entropy-21-00040-t005:** Sensitivity of the proposed streaming scheme.

File Name	Chaotic Map		Plaintext		
	b	*X* _0_	0.25	0.5	0.75
Xml	48.60%	48.63%	48.59%	48.54%	49.31%
Ooffice	49.76%	49.72%	49.69%	49.60%	49.76%
Reymont	49.42%	49.45%	49.32%	49.18%	49.00%
Sao	49.83%	49.83%	49.90%	49.91%	48.36%
X-ray	49.82%	49.77%	49.74%	49.41%	49.06%
Mr	49.19%	49.26%	49.25%	49.35%	49.10%
Osdb	49.80%	49.81%	49.81%	49.79%	49.79%
Dickens	49.08%	49.08%	49.16%	49.15%	49.17%
Samba	48.82%	48.87%	49.58%	48.96%	49.34%
Nci	46.56%	46.54%	46.58%	46.55%	46.80%

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
