# Peer review of "Three Techniques for Enhancing Chaos-Based Joint Compression and Encryption Schemes"

_entropy, 2019, doi:10.3390/e21010040_

Round 1

Reviewer 1 Report

Dear authors,

Your work makes sense but very major revisions are required to highlight your contribution. Hopefully, you can attend all these recommendations to publish a nice article.

You are encouraged to review english, for example in line 442 In this work, the compression ratio is computes ELIMINATE is OR computed

You need to include details on your implementation and revise the following issues:

line 39 The receiver is able to synchronize the changed statistical model and produce the correct output only when the secret key is provided, correctly.

You can discuss on synchronizing techniques that has been applied to secure communications

line 47 The prominent properties of chaotic systems, such as sensitive to initial conditions, topologically mixing, and having dense periodic orbits, are ideal for building cryptography schemes

This is a good property of chaotic systems, in this part you can discuss that chaotic maps are iterative formulae compared to continuous time chaotic systems, which require to establish the correct step size and numerical method to implement then in digital hardware, as demonstrated in the key reference: FPGA-based implementation of chaotic oscillators by applying the numerical method based on trigonometric polynomials

AD Pano-Azucena, E Tlelo-Cuautle, G Rodriguez-Gomez, LG de la Fraga

AIP Advances 8 (7), 075217, 2018.

In the following subsection, you must eliminate the references are the end of each sentence. The references are mentioned in the main text and neither in the abstract nor in the sections/subsections titles!

2.1. Look-up Table and Chaotic-map based Approach [8]

2.2. Dynamic Updating Look-up-table based Approach [9]

2.3. Number of Distinct Plaintext Symbol based Approach [10]

2.4. Bi-gram based Approach [11]

2.5. JACE Scheme for Streaming Data [12]

line 136 We first present three new techniques that respectively improve the JCAE schemes presented in [9], [11] and [12]. IN THIS CASE YOU MUST ELIMINATE REFERENCE [11] because a Thesis even for graduate studies, is not available as a journal article is. You must search recent related literature to change this reference and then to add new results that can improve your contribution.

9. J. Chen, J. Zhou and K. W. Wong, “A modified chaos-based joint compression and encryption scheme,”  IEEE Trans. Circuits Syst. II, Exp. Briefs, vol. 58, no. 2, pp.110–114, 2011.

11. Yu-Chen Lin and Ja-Ling Wu, “A novel chaos-based joint compression and encryption scheme using normalized conditional bi-gram probability,” Master Thesis, Graduate Institute of Computer Science and Information Engineering, National Taiwan University, 2016.

12. Yu-Jung Chang, Yu-Chen Lin, Yung-Chen Hsieh, Chih-Wen Hsueh, and Ja-Ling Wu, “A chaos-based joint compression and encryption scheme for streaming data,” The IEEE 14th Conference on Advanced Trusted Computing, August 2017.

Figure 3.1. The block diagram of the proposed static scheme. THIs FIGURE Highlights masking, which has demonstrated to be a good option in encryption schemes implemented in FPGAs, see for example the key reference:

FPGA-based Chaotic Cryptosystem by Using Voice Recognition as Access Key, E Rodríguez-Orozco, E García-Guerrero, E Inzunza-Gonzalez, ...

Electronics 7 (12), 414 2018

Table 4.1. Performance comparisons of the static schemes. IN THIS TABLE YOU MUST DELETE REFERENCE [11], as mentioned above, citing journal articles of recent years is more adequate than citing graduate theses, which are difficult to verify if the results are really good.

line 487 Chapter 5 Conclusion...In this case you must delete Chapter

Finally, you can discuss hardware implementations of chaotic maps, which speed is higher than software realizations,

IN THE SAME LINE FOR RESEARCH ARE THE INTEGRATED DESIGNS OF CHAOTIC OSCILLATORS, you can discuss their usefulness in the internet of things due to its very low power consumption and that they can be embedded into any system because the integrated circuit technology is in the scale of nanometers, and the oscillators can be optimized to provide better security, see the reference

Optimization and CMOS design of chaotic oscillators robust to PVT variations, VH Carbajal-Gomez, E Tlelo-Cuautle, JM Muñoz-Pacheco, ...

Integration the VLSI journal, 2018

Author Response

Our comments are in yellow color.

Comment-1. Your work makes sense but very major revisions are required to highlight your contribution. Hopefully, you can attend all these recommendations to publish a nice article.

Thanks for the encouragement, we will try our best to complete our revision within the deadline.

Comment-2. You are encouraged to review English, for example in line 442 In this work, the compression ratio is computes ELIMINATE is OR computed

As suggested, we have corrected the typo from “computes” to “computed”.

Comment-3. You need to include details on your implementation and revise the following issues: line 39 The receiver is able to synchronize the changed statistical model and produce the correct output only when the secret key is provided, correctly. You can discuss on synchronizing techniques that has been applied to secure communications

Actually, in line 39,our statement about “The receiver is able to synchronize the changed statistical model and , …,” is a general comment to the characteristics of the approach for embedding key-controlled confusion and diffusion mechanisms into compression algorithms. In order to make the target of our statement more clear, we add the related references [1-3] at the end our statement.

Comment.4. line 47 The prominent properties of chaotic systems, such as sensitive to initial conditions, topologically mixing, and having dense periodic orbits, are ideal for building cryptography schemes. This is a good property of chaotic systems, in this part you can discuss that chaotic maps are iterative formulae compared to continuous time chaotic systems, which require to establish the correct step size and numerical method to implement then in digital hardware, as demonstrated in the key reference: FPGA-based implementation of chaotic oscillators by applying the numerical method based on trigonometric polynomials --- A. D. Pano-Azucena, E. Tlelo-Cuautle, G. Rodriguez-Gomez, L. G. de la Fraga AIP Advances 8 (7), 075217, 2018.

As suggested, we added a paragraph to emphasize the differences between chaotic maps and continuous time chaotic systems, specifically in hardware implementation. Moreover, a new reference is added to support this remark. Thanks to the anonymous reviewer for bringing the reference to our attention!

Comment-5. In the following subsection, you must eliminate the references are the end of each sentence. The references are mentioned in the main text and neither in the abstract nor in the sections/subsections titles!

 2.1. Look-up Table and Chaotic-map based Approach [8]

 2.2. Dynamic Updating Look-up-table based Approach [9]

 2.3. Number of Distinct Plaintext Symbol based Approach [10]

 2.4. Bi-gram based Approach [11]

 2.5. JACE Scheme for Streaming Data [12]

As suggested, all the references appeared in the sections/subsections titles are removed.

Comment-6. line 136 We first present three new techniques that respectively improve the JCAE schemes presented in [9], [11] and [12]. IN THIS CASE YOU MUST ELIMINATE REFERENCE [11] because a Thesis even for graduate studies, is not available as a journal article is. You must search recent related literature to change this reference and then to add new results that can improve your contribution.

[9]. J. Chen, J. Zhou and K. W. Wong, “A modified chaos-based joint compression and encryption scheme,” IEEE Trans. Circuits Syst. II, Exp. Briefs, vol. 58, no. 2, pp.110–114, 2011.

[11]. Yu-Chen Lin and Ja-Ling Wu, “A novel chaos-based joint compression and encryption scheme using normalized conditional bi-gram probability,” Master Thesis, Graduate Institute of Computer Science and Information Engineering, National Taiwan University, 2016.

[12]. Yu-Jung Chang, Yu-Chen Lin, Yung-Chen Hsieh, Chih-Wen Hsueh, and Ja-Ling Wu, “A chaos-based joint compression and encryption scheme for streaming data,” The IEEE 14th Conference on Advanced Trusted Computing, August 2017.

We do agree the reviewer’s comment about the availability of a master degree Thesis of a graduate institute; however, reference [11] is a necessary intermedium for explaining how reference [12] is evolved from [8] to [10]. For releasing this conflict, a paper version of [11], which is searchable by any search engine from the Internet, is used to replace the original reference. Through the attached web-address, we do believe the raised availability issue can be much released.

Comment-7. Figure 3.1. The block diagram of the proposed static scheme. THIs FIGURE Highlights masking, which has demonstrated to be a good option in encryption schemes implemented in FPGAs, see for example the key reference:

FPGA-based Chaotic Cryptosystem by Using Voice Recognition as Access Key, E Rodríguez-Orozco, E García-Guerrero, E Inzunza-Gonzalez, ...Electronics 7 (12), 414 2018

As suggested, a paragraph is added in the bottom of Fig. 3.1 to highlight the merit of realizing the masking module by FPGA. Moreover, a new reference is added to support this remark.Thanks to the anonymous reviewer for bringing the reference to our attention!

Comment-8. Table 4.1. Performance comparisons of the static schemes. IN THIS TABLE YOU MUST DELETE REFERENCE [11], as mentioned above, citing journal articles of recent years is more adequate than citing graduate theses, which are difficult to verify if the results are really good.

This is a follow-up comment of Comment-6. Since the availability issue of reference [11] have been smoothed as mentioned in our reply to Comment-6. The concern raised in this comment has been much released!

Comment-9. line 487 Chapter 5 Conclusion...In this case you must delete Chapter

We have deleted the word “Chapter”, as suggested.

Comment-10. Finally, you can discuss hardware implementations of chaotic maps, which speed is higher than software realizations,

IN THE SAME LINE FOR RESEARCH ARE THE INTEGRATED DESIGNS OF CHAOTIC OSCILLATORS, you can discuss their usefulness in the internet of things due to its very low power consumption and that they can be embedded into any system because the integrated circuit technology is in the scale of nanometers, and the oscillators can be optimized to provide better security, see the reference

Optimization and CMOS design of chaotic oscillators robust to PVT variations, V. H. Carbajal-Gomez, E. Tlelo-Cuautle, J. M. Muñoz-Pacheco, ..., Integration the VLSI journal, 2018

To tell the truth, all of us are majored in Computer Science and not so familiar with the newest integrated circuit technology, such as the new progress in CMOS design for chaotic oscillators. However, we do agree reviewers’ comment about the superiority in using hardware to implement chaotic maps. Therefore, a paragraph is added at the end of the Conclusion section to acknowledge this fact. Similarly,a new reference is added to support this remark.Thanks to the anonymous reviewer for bringing the reference to our attention!

Reviewer 2 Report

the authors must to add research about linear complexity, key size analysis, package statistical test of Diehard and ENT.

I want to see comparative tables with other chaotic based  pseudorandom generators, for example 

DOI: 10.1063/1.4758983

DOI: 10.1088/1674-1056/21/9/090506

DOI: 10.12988/astp.2015.5342

DOI: 10.1007/s10836-018-5767-0

Author Response

Comment-1. The authors must to add research about linear complexity, key size analysis, package statistical test of Diehard and ENT.

Our reply to comment-1.

According to the references recommended by reviewer 2, we do think he or she is an expert highly interested in the Statistical Properties of Chaotic Cryptographic Schemes generated by various Chaotic equations. This explains why reviewer 2 asking for adding researches about linear complexity and key size analysis on Diehard and ENT datasets. Unfortunately, we didn’t fulfill this requirement in our revision, due to the following reasons:

a.    The Chaotic-map based Random Number Generatoris only one of several modules of our Joint Compression and Encryption Scheme (as shown in Figs. 3.1 and 3.2);therefore, system level performance metricssuch as: Compression RatioExecution Speed, and System Security (reflected by key sensitivity and plaintext sensitivity, given in Section 4.3) are the chosen performance metrics for evaluating all our benchmarking works. On the other hand, linear complexity and key size analysis are parts of the NIST Statistic Test Suite, which is usually used to measure randomness of the binary sequence generated by a chaotic cryptographic system

Moreover, from the definitions of Diehardand ENT(Wikipedia https://en.wikipedia.org/wiki/Diehard_tests : Diehard - published byProfessor George Marsaglia in 1995, as part of the Marsaglia Random Number CDROM ; ENT - The Fourmilab ENTprogram is a public domain utility which tests binary data sequences, either as a series of 8 bit bytes, or as a bit stream, with five standard tests for randomness)package statistical tests of them cannot truly reflect our system performances.

b.     Thanks to reviewer 2 for bringing the issues of Diehard and ENT Statistical Testsfor various Chaotic maps (such as those will be addressed in 2) to our attention. However, the focusesof those two tests are not so related to our work, and also due to tight revision time-limitand lacking of experimental comparison results in our benchmark works, we didn’t respond to this comment in our revision. 

Comment-2. I want to see comparative tables with other chaotic based pseudorandom generators, for example 

DOI: 10.1063/1.4758983, DOI: 10.1088/1674-1056/21/9/090506, DOI: 10.12988/astp.2015.5342, DOI: 10.1007/s10836-018-5767-0.

Our reply to Comment-2.

Thanks to reviewer 2 for brining the above four papers to our attention! For the ease of referencing, we denote the above 4 papers respectively as :

DOI: 10.1063/1.4758983 [Ref.1], DOI: 10.1088/1674-1056/21/9/090506 [Ref 2], DOI: 10.12988/astp.2015.5342 [Ref. 3], and DOI: 10.1007/s10836-018-5767-0 [Ref.4].

Let’s briefly summarize the contributions of each one of the above writeups in the following, first

[Ref.1] Borislav P. Stoyanov, “Chaotic Cryptographic Scheme and its Randomness Evaluation,” AIPConference Proceedings · October 2012.

[Ref. 1] proposed a new cryptographic scheme based on theLorenz Chaos Attractor(which consists of three ordinary differential equations) and 32-bit bent Boolean function. The keystream generated by the scheme was evaluated with batteries of the NIST statistical tests; therefore, p-valueand pass rate are the chosen metrics for evaluating system performances. It was shown in [Ref. 1] that the new cryptographic scheme ensures a secure way for sending digital data with potential applications inreal-time image encryption.Moreover, the NIST statistical test suite was used to measure the randomness of the binary sequence generated by the proposed chaotic cryptographic system, andthe first 8 color images of the USC-SIPI image database are taken as the benchmarking images. The security analysis results showed that the proposed chaotic pseudorandom system can assure the required security in digital communications.

[Ref. 2] Liu Yang and Tong Xiao-Jun, “A new pseudorandom number generator based on a complex number chaotic equation, “Chinese Phys. B, Vol. 21, No. 9 (2012). 

[Ref. 2] proposed and proved that the equation xn+1=x 2+ 1/ (2 xn) ischaotic in the imaginary axis. That is, the definitional domain of the chaotic equation is generalized from the real number field to the complex one, and a pseudorandom number generator and the corresponding binary sequences are constructed, accordingly. Once again, in the experiments, NIST test suite was used to detect deviations of a binary sequence from the true randomness. For each test, a P value is computed from the binary sequence. If this value is greater than a predefined threshold α, it is considered that the sequence passes the test successfully. Both theoretical analysis and experimental results showed that the randomness, the security, and the speed of the binary sequences generation by the pseudorandom number generator are satisfactory. 

[Ref.3] Borislav Stoyanov, “Novel Secure Pseudo-Random Number Generation Scheme Based on Two Tinkerbell Maps,” Advanced Studies in Theoretical Physics Vol. 9, 2015, no. 9, 411 – 421.

[Ref. 3] proposed a Tinkerbell map as a novel pseudo- random number generator. The proposed approach was evaluated with various statistical packages: NIST, DIEHARD and ENT. The results of the analysis demonstrate that the new derivative bit stream scheme is very suitable for embedding in critical cryptographic applications, where the Tinkerbell map is a two-dimensional discrete-time dynamical system given by: 

xn+1 =x2n− yn+a x+b y;   yn+1 = 2 xy+ c x+ d y,                          where a = 0.9, b = −0.6013, c = 2.0 and d = 0.50. And NIST, DIEHARD, and ENT statistical test suites were used as the testing benchmark, again. The statistical tests analysis results demonstrated that the new algorithm can assure high level of pseudo- randomness in critical cryptographic applications.

[Ref. 4] Dragan Lambic ́,AleksandarJankovic ́, MusheerAhmad, “Security Analysis of the Efficient Chaos Pseudo-Random Number Generator Applied to Video Encryption,” Journalof Electronic Testing (2018) 34:709–715

Security analyses of an existing chaos pseudo-random number generator, which had been applied to video encryption, was presented in [Ref. 4]. Analysis of chaotic maps used in the analyzed PRNG (which is constructed based on the following two chaotic maps: xn+1=2xn2-1 and yn+1=4yn3– 3yn, where x, y[-1,1]) showed that these chaotic maps do not enhance security of the analyzed PRNG due to the fact that considerable number of initial values lead to fixed points. Also, based on 6 known iterations of the analyzed PRNG, data sufficient for reconstruction of whole pseudo random sequence can be obtained with complexity which is much smaller than the estimated key space. Therefore, authors of [Ref.4] remarked that the security of the analyzed PRNG is much lower than expected and it should be used with caution. Finally, some potential improvements of the analyzed PRNG were proposed which could eliminate perceived shortcomings of the original version.

From the above summarization, we found that the first three referred papers focused on checking the security and/or randomness properties of the binary sequences generated from PRNGs constructed by some newly proposed chaotic maps ; therefore, NIST, DIEHARD, and ENT statistic tests are conducted to verify the superiority of their work. The authors of the fourth paper launched an effective attack to a published Chaos PRGN and showed thatthe security of the analyzed PRNG is much lower than expected and it should be used with caution.

Based on the above discussions, our rely to this comment includes the following 3 points:

1.    As stated in our rely to the first comment, the Chaotic-map based PRNGis only one of the modules of our Joint Compression and Encryption Scheme and the provided performance evaluation metrics of all referred papers are not the same as those adopted in our work, “ Including comparative tables with other chaotic based pseudorandom generators” in our work seems not so reasonable and necessary.

2.    On the other hand, we understand and acknowledge that for those who are working on new Chaotic-map based PRGN, both comment-1 and comment-2 are important issues that should be taken into consideration. Actually, we do interest in understanding the changes of our system security measures and overall system performances if our Logistic-map based PRGN module were replaced by other PRGN modules based on different Chaotic-maps; however, lots of experiments and comparisons have to be conducted before a formal conclusion can be reached. It seems not so meaningful, to us, to devote so many times and resources in doing these tasks in this revision.

3.    We do appreciate reviewer 2 for brining up these two valuable comments and the four referred papers to us. Although we didn’t take direct responding actions to them, we do think the above-mentioned topics are worthy of further exploring in the future. Therefore, a paragraph is added in the conclusion section as our responses to these comments.

Reviewer 3 Report

The paper provides three methods for enhancing chaos-based joint compression and encryption schemes. The work is very interesting, well organized and easy to follow. However, some minor issues have to be solved:

a) the affiliations of authors are not provided in the manuscript;

b) the last sentence from abstract (lines 17-19) is unclear;

c) the sentence started in line 26 ("It follows...") is badly shaped and needs to be polished.

d) line 180: instead of "skip and go" it is written "skip and goes";

e) in eq. from line 347 is an unwanted space between 3 and 2;

f) line 430: it is written "Chapter 4" instead of simply "4." ; the same for line 487;

g) line 461: a reference for logistic map is needed;

h) the section "Author contributions" required by journal's Information for Authors is missing.

g) the references must be formatted according to journal's Information for Authors.

Author Response

1.    The paper provides three methods for enhancing chaos-based joint compression and encryption schemes. The work is very interesting, well organized and easy to follow. 

Thanks to reviewer 3 for his/her encouragement, we will try our best to complete the revision before the deadline.

2.    However, some minor issues have to be solved:

a)    the affiliations of authors are not provided in the manuscript; 

We have added “the affiliations of authors” in the revision, as suggested.

b) the last sentence from abstract (lines 17-19) is unclear;

As suggested, we have slightly modified our abstract to clarify our meaning as possible as we can.

c) the sentence started in line 26 ("It follows...") is badly shaped and needs to be polished.

As suggested, we have changed the marked sentence so as to clarify our meaning as possible as we can.

d) line 180: instead of "skip and go" it is written "skip and goes";

We have changed the above sentence as suggested.

e) in eq. from line 347 is an unwanted space between 3 and 2;

We have removed the unwanted spacing, as suggested.

f) line 430: it is written "Chapter 4" instead of simply "4." ; the same for line 487;

The word “Chapter” has been removed from the title of Sections 4 and 5, as suggested.

g) line 461: a reference for logistic map is needed;

The following reference has been added in the revision, as suggested. [17] M. Ausloos, M. Dirickx, “The logistic map and the route to chaos from the beginnings to modern applications,” Springer, 2006.

h) the section "Author contributions" required by journal's Information for Authors is missing.

We have added “Author contributions” in the revision, as suggested.

i) the references must be formatted according to journal's Information for Authors.

We have re-formatted the references, as suggested.

General comments:

1. Please provide your *detailed responses* point-by-point to the reviewers' comments. Please include in your rebuttal if you found it impossible to address certain comments. The revised version will be inspected by the editors and reviewers.

We have provided our *detailed responses* point-by-point to the reviewers' comments, in the revision, as suggested.

2.    Any revisions should be *clearly highlighted*, for example using the 
"Track Changes" function in Microsoft Word, so that changes are easily visible to the editors and reviewers. 

We did it, as suggested!

3. Please replace the email addresses ([email protected]/[email protected]) 
with affiliation email addresses.

We did it, as suggested.

Round 2

Reviewer 1 Report

The updated version is fine to be accepted

Reviewer 2 Report

All of my comments are done